# Ca^2+^ Signalling Differentially Regulates Germ-Tube Formation and Cell Fusion in *Fusarium oxysporum*

**DOI:** 10.3390/jof8010090

**Published:** 2022-01-17

**Authors:** Smija M. Kurian, Alexander Lichius, Nick D. Read

**Affiliations:** 1Manchester Fungal Infection Group, University of Manchester, Manchester M13 9NT, UK; nick.read@manchester.ac.uk; 2College of Life and Environmental Sciences, University of Exeter, Exeter EX4 4QD, UK; 3Department of Microbiology, University of Innsbruck, 6020 Innsbruck, Austria; alexander.lichius@uibk.ac.at

**Keywords:** Ca^2+^ signalling, cell fusion, CAT, germination, *Fusarium oxysporum*

## Abstract

*Fusarium oxysporum* is an important plant pathogen and an emerging opportunistic human pathogen. Germination of conidial spores and their fusion via conidial anastomosis tubes (CATs) are significant events during colony establishment in culture and on host plants and, hence, very likely on human epithelia. CAT fusion exhibited by conidial germlings of Fusarium species has been postulated to facilitate mitotic recombination, leading to heterokaryon formation and strains with varied genotypes and potentially increased virulence. Ca^2+^ signalling is key to many of the important physiological processes in filamentous fungi. Here, we tested pharmacological agents with defined modes of action in modulation of the mammalian Ca^2+^ signalling machinery for their effect on germination and CAT-mediated cell fusion in *F. oxysporum*. We found various drug-specific and dose-dependent effects. Inhibition of calcineurin by FK506 or cyclosporin A, as well as chelation of extracellular Ca^2+^ by BAPTA, exclusively inhibit CAT induction but not germ-tube formation. On the other hand, inhibition of Ca2+ channels by verapamil, calmodulin inhibition by calmidazolium, and inhibition of mitochondrial calcium uniporters by RU360 inhibited both CAT induction and germ-tube formation. Thapsigargin, an inhibitor of mammalian sarco/endoplasmic reticulum Ca^2+^ ATPase (SERCA), partially inhibited CAT induction but had no effect on germ-tube formation. These results provide initial evidence for morphologically defining roles of Ca^2+^-signalling components in the early developmental stages of *F. oxysporum* colony establishment—most notably, the indication that calcium ions act as self-signalling molecules in this process. Our findings contribute an important first step towards the identification of Ca^2+^ inhibitors with fungas-specific effects that could be exploited for the treatment of infected plants and humans.

## 1. Introduction

Ca^2+^ signalling is known to play a major role in regulating important morphogenetic and physiological processes in filamentous fungi. Evidence has been obtained suggesting the involvement of Ca^2+^ in the regulation of morphological and pathogenic events in fungi, such as secretion [1,2], hyphal tip growth [1,3,4], hyphal branching [5,6], sporulation [7,8], dimorphism [9], cytoskeletal organization [1] and differentiation of infection structures [10]. As many drugs in humans target Ca^2+^-signalling machinery, deciphering the role of Ca^2+^ signalling in morphological and physiological changes in fungi can help to unravel their potential as antifungal drug targets.

Fungal spores serve as dissemination structures that can survive harsh environmental conditions outside the host. Spore germination, with cell-symmetry breaking through the emergence of a germ tube and/or CAT as the initiating process, is the first step in the life cycle of most filamentous fungi, including the economically important fungal pathogen *Fusarium oxysporum*. *F. oxysporum* produces three different types of vegetative, (i.e., asexual) spores: microconidia, macroconidia and chlamydospores [11]. Upon germination, microconidia of *F. oxysporum* undergo cell-cell fusion via conidial anastomosis tubes (CATs), which has been postulated to promote efficient colonization and horizontal gene transfer (HGT), leading to heterokaryon formation [12,13]. The process of HGT is important, as it can lead to strains with acquired or altered virulence and resistance [14]. This is particularly important in the context of acquiring resistance to antifungal drugs. The process of CAT-mediated cell fusion progresses in three stages: induction of CAT formation, CAT homing and CAT fusion [13]. Almost 60 mutants involved in different signalling pathways have been identified as defective at different stages of CAT fusion in *Neurospora crassa* [15]. In this model fungus, preliminary evidence was obtained for the role of Ca^2+^ signalling during CAT induction. The oscillatory recruitment of proteins MAK-2 and SO to CAT tips during CAT fusion was predicted from mathematical modelling to involve the pulsatile secretion of a self-signalling chemo-attractive ligand, which is not yet identified [16]. This pulsatile secretion and the merging of the plasma membranes of fusing CATs have been hypothesized to be regulated by Ca^2+^ signalling [17].

In fungi, there are two types of Ca^2+^ uptake systems involving Ca^2+^ channel transport across the plasma membrane: the low-affinity Ca^2+^ uptake system (LACS) and the high-affinity Ca^2+^ uptake system (HACS). The L-type Ca^2+^ channel Cch1 and regulatory proteins Mid1 and Ecm7 [18,19] belong to HACS. Cch1 has been as the mammalian orthologue of the pore-forming α1 subunit of voltage-gated calcium channels (VGCCs), although there is only 24% amino-acid sequence similarity between the two [20,21]. The HACS operates when the availability of external Ca^2+^ is low, whilst the LACS functions when there is a high availability of Ca^2+^ in the external medium. The only known member of LACS is Fig1 [22,23]. Upon stimulation, extracellular Ca^2+^ uptake takes place through either low-affinity or high-affinity Ca^2+^ channels in the plasma membrane. This increases the resting concentration of Ca^2+^ (50–200 nm in the case of most fungal cells) in the cytoplasm by around 1000-fold [2]. Opening of Ca^2+^ storage organelles, such as ER, Golgi, mitochondria and vacuoles, also contributes to the increase in intracellular Ca^2+^. This increase remains transient in cells due to sequestration of Ca^2+^ within organelles and/or as a result of its export by the action of Ca^2+^ antiporters and Ca^2+^ pumps. Meanwhile, the transient increase in intracellular [Ca^2+^]_c_ forms a Ca^2+^ signal, which activates downstream signalling pathways through Ca^2+^-dependent proteins, such as calmodulin. Proteins such as calmodulin undergo conformational changes upon binding to four Ca^2+^ ions each [24]. Binding of the Ca^2+^-calmodulin complex in turn activates phosphatases, such as calcineurin, which regulates numerous downstream physiological processes and morphological changes through dephosphorylation of transcription factors, such as the well characterized Crz1 [2]. In mammalian cells, stimulus recognition through GPCRs at the plasma membrane leads to generation of inositol triphosphate (IP_3_). Binding of IP_3_ to its receptor, IP_3_R, in the ER releases Ca^2+^ into the cytoplasm. The ER membrane also contains ryanodine receptors (RyRs), which are opened by Ca^2+^ itself, leading to calcium-induced calcium release (CICR). The efflux of Ca^2+^ to the cytoplasm is followed by its sequestration to the ER through the Ca^2+^ ATPase pump [25].

Besides the use of mutants and the direct imaging/measurement of [Ca^2+^]_c_ dynamics in living cells, pharmacological agents that modulate different components of the Ca^2+^-signalling machinery in animal cells have been extensively employed to study the role of Ca^2+^ signalling in filamentous fungi (e.g., [26,27]. The mode of action of most of these Ca^2+^ modulators have been well characterized in mammalian cells. 1,2-bis(o-aminophenoxy)ethane-*N*,*N*,*N*′,*N*′-tetraacetic acid (BAPTA) chelates extracellular Ca^2+^ and inhibits the availability of extracellular Ca^2+^ for influx to cells [28,29]. Verapamil inhibits Ca^2+^ channels in the plasma membrane and thus inhibits the influx of Ca^2+^ [30,31]. Calmidazolium blocks calmodulin and prevents binding of intracellular Ca^2+^ to calmodulin after a transient Ca^2+^ increase in the cytoplasm [2,24,32]. This prevents further downstream signalling upon stimulus recognition on the plasma membrane. FK506 (aka tacrolimus) [33] binds to FK506 binding proteins (FKBPs), which in turn bind to calcineurin and prevent its activation by the Ca^2+^-calmodulin complex [34,35,36]. An alternate inhibitory path exists for calcineurin. Cyclosporin A, a cyclic polypeptide, forms a complex with cyclophilin and inactivates calcineurin function [37]. Both these inhibitory pathways prevent the phosphatase activity of calcineurin, which otherwise dephosphorylates downstream transcription factors, affecting altered gene expression in response to Ca^2+^ signalling. In mammalian cells, thapsigargin binds irreversibly and specifically to F256 in the M3 helix transmembrane domain of the sarco/endoplasmic reticulum Ca^2+^ ATPase (SERCA) pump on the ER membrane, leading to inhibition of Ca^2+^ influx to the ER lumen [27,38,39]. The mitochondrial calcium uniporter (MCU) in the mitochondrial inner membrane allows for sequestration of Ca^2+^ to the mitochondria following a Ca^2+^ signal in the cytoplasm [40,41,42]. Ruthenium red (RU360) has been shown to have an inhibitory effect on MCU [27].

Comparative genomic analyses and functional screening of components of the Ca^2+^-signalling machinery of animals and plants has revealed the conservation of numerous components in different filamentous fungi, including *Fusarium oxysporum* [8,43,44,45]. However, some important components have been shown to not be conserved [43,46]. Homologous sequence analysis of cation-signalling components to mammalian protein sequences revealed the absence of IP_3_R and RYR homologues in all fungi examined [47]. However, similar channels are present in the vacuole, which is a major Ca^2+^-storage organelle in fungi. The MCU also has homologues in filamentous fungi, including *N. crassa*, *Aspergillus* spp. and *Fusarium* spp. [27,47,48]. ER-localized SERCA-type Ca^2+^ -ATPases, Nca1/Eca1, Ca^2+^/Mn^2+^ -ATPase and Spf1 have been characterized in other filamentous fungi, although they have not yet been studied in Fusarium [8].

Although *F. oxysporum* is a major plant pathogen and an opportunistic human pathogen [49,50,51], there are very few studies on the role of Ca^2+^ signalling in this important species. Inhibition of calcineurin function, both through creation of respective deletion strains and by pharmacological inhibition using FK506 and Cyclosporin A, confirmed a role for calcineurin in germination of microconidia in *F. oxysporum* under rich media conditions [3,52]. Here, we used known Ca^2+^-signalling modulators to study the role of the Ca^2+^-signalling machinery in germ-tube formation and CAT-mediated cell fusion of *F. oxysporum* f. sp. *Lycopersici* (*Fol*) in order to report the first evidence concerning how inhibition of key Ca^2+^ signalling components affects the early stages of germling development in this fungal pathogen. New insights on this topic will hopefully contribute to the identification of Ca^2+^ inhibitors with fungus-specific effects that could be exploited for the treatment of infected plants and humans.

## 2. Materials and Methods

### 2.1. Culture Conditions of F. oxysporum

Microconidia of *F. oxysporum* f. sp. *Lycopersici* (isolate 4287) were harvested from liquid cultures grown at 25 °C and 250 rpm for 5 days in PDB [13]. Germination and CAT fusion assays were set up in 8-well borosilicate slide-culture chambers (Nalgene Nunc, Rochester, NY, USA), as previously described in detail [13]. Briefly, 300 µL of a 1 × 10^6^ cells/mL spore suspension in media was added to each well and incubated for 12 h at 25 °C. Spore germination was analysed in both 1% PDB alone and in 1% PDB supplemented with 25 mM NaNO_3_. NaNO_3_ is required for CAT-mediated germling fusion to occur in *F. oxysporum*. Consequently, CAT induction and fusion were only analysed in the NaNO_3_-supplemented medium because the process is inhibited in the 1% PDB control condition. We have reported the absence of CAT fusion as an inhibition of the first stage in CAT-mediated cell fusion—which is CAT induction/CAT formation—because we did not find any condition involving the tested pharmacological agents in which CATs were formed but were unable to fuse.

### 2.2. Pharmacological Inhibition, Microscopy and Statistical Analysis

Pharmacological agents applied to assess the influence of Ca^2+^ inhibition on spore germination and CAT fusion are shown in Table 1. Stocks of the agents were prepared with either water or DMSO. An appropriate volume of each inhibitor stock solution was added to the growth medium (either 1% PDB or 1% PDB + 25 mM NaNO_3_) at the beginning of each experiment to provide the desired final concentration. Effects of these agents on germ-tube formation was analysed by cultivating microconidia in 1% PDB alone, in which only germ tubes were being formed, while the effect on CAT-mediated cell fusion was analysed by cultivation in fusion medium, i.e., 1% PDB supplemented with 25 mM NaNO_3_, in which both germ tubes and CATs were being formed. Unless otherwise stated, the final concentration of DMSO did not exceed 4%.

The formation of germ tubes and/or CATs was determined at 12 h post incubation (hpi) using simple DIC light microscopy and subsequent image quantification, as detailed by Kurian et al. (2018). A total of 360 imaging files, with each comprising a 4 × 4 image array, were collected for this study. From this collection of 5000 images, a minimum of 20 images were selected to morphologically evaluate at least 300 cells per test condition per experiment. Results were assembled from a minimum of three experimental repeats for each drug concentration tested, and the mean ± SEM was plotted. Plotting of graphs and performance of statistical analysis were performed using GraphPad Prism version 8.0.0 for Windows (GraphPad Software, San Diego, CA, USA, www.graphpad.com (accessed on 25 August 2021)). Non-parametric Mann-Whitney tests were conducted for the comparison of results from each drug concentration versus non-treated, and *p*-values ≤ 0.05 were reported as significant (*).

The minimum inhibitory concentration (MIC) is the concentration of a pharmacological agent resulting in ≥90% inhibition of its target molecule and process in comparison to the untreated control. MIC values were determined for germ-tube formation from the counts of germ tubes being formed, as well as non-germinated spores, upon treatment with a specified concentration of an inhibitor. Similarly, MIC values for CAT fusion were determined from counts of germinated spores undergoing CAT fusion upon treatment with specified concentrations of the same inhibitors.

## 3. Results

The dose-dependent inhibitory effects of the Ca^2+^ modulators BAPTA, verapamil, calmidazolium, thapsigargin, FK506 and RU360 on microconidial germ-tube formation and CAT-mediated germling fusion were quantitatively analysed (Figure 1). The tested concentration ranges of these inhibitors all fell within those previously reported to have inhibitory effects in fungi [34,48].

### 3.1. Extracellular Ca^2+^ Is Required for CAT Induction but Not Germ-Tube Formation

We used BAPTA, which is cell-impermeable and hence chelates extracellular Ca^2+^, at concentrations between 0.5 and 15 mM to study the role of extracellular Ca^2+^ in conidial germ-tube formation and CAT fusion (Table 1 and Figure 2). Germ-tube formation in 1% PDB alone was unaffected by BAPTA concentrations of up to 5 mM but showed up to 10% reduction with 10–15 mM BAPTA. Germ-tube formation in fusion medium (1% PDB + NaNO_3_) was not inhibited by BAPTA at concentrations of up to 15 mM. However, CAT induction was very sensitive to BAPTA, and the MIC value of BAPTA for CAT fusion was determined as 5 mM. As BAPTA is a highly specific chelating agent compared to other less effective agents, such as ethylene glycol bis (2-Aminoethyl ether)-*N*,*N*,*N*′,*N*′ tetraacetic acid (EGTA), we assume complete removal of the available Ca^2+^ from the medium at 1 mM of BAPTA [53]. It is evident from these results that conidial germ-tube formation in *Fol* is still functional in the absence of extracellular Ca^2+^, either due to intracellular Ca^2+^ from storage organelles or via compensation by other extracellular ions. CAT induction, on the other hand, is strictly dependent on extracellular Ca^2+^, most probably because it is involved in the cell-cell communication between the fusing partners.

### 3.2. Calcium Entry/Exit through Calcium Channels Is Required for Germ-Tube Formation and CAT Induction

Verapamil, an inhibitor of the CCH1 Ca^2+^ channel, was used at concentrations between 0.5 and 15 mM to determine the role of Ca^2+^ influx through the plasma membrane in spore germ-tube formation and CAT fusion (Table 1 and Figure 2). Cell-symmetry breaking by germ-tube formation alone in 1% PDB (w/o NaNO_3_) was inhibited by 5 mM verapamil (MIC = 5 mM). Cell-symmetry breaking by germ-tube formation in fusion medium (1% PDB + NaNO_3_) showed a similar dose-dependent inhibition but required 10 mM verapamil for inhibition (MIC = 10 mM). The presence of additional nutrient ions from NaNO_3_ are potentially responsible for this 5 mM concentration gap until complete inhibition between both media. CAT fusion responded more sensitively than germ-tube formation and was already fully inhibited by 2 mM verapamil in fusion medium (MIC = 2 mM). This shows that both CAT induction and germ-tube formation require an influx of extracellular Ca^2+^ through the CCH1 channel, albeit at different concentrations. As CCH1 is a component of HACS, which functions under low extracellular Ca^2+^ concentrations, our results indicate the requirement of a higher extracellular Ca^2+^ concentration for CAT induction. We suspect that a complete block of CCH1 by verapamil does not occur at the inhibitory concentration for CAT induction. A lower uptake of extracellular Ca^2+^ might still take place through the CCH1 channel, which might not be enough to induce CAT formation but could be sufficient for germ-tube formation. Complete inhibition of CCH1 might be caused by the inhibitory concentration of 10 mM verapamil for germ-tube formation.

### 3.3. Calmodulin Inhibition by Calmidazolium Inhibits CAT Induction and Germ-Tube Formation

The calmodulin inhibitor calmidazolium was used at concentrations between 0.5 and 20 µM to determine the function of intracellular Ca^2+^ in germ-tube formation and CAT fusion (Table 1 and Figure 2). The MIC value of calmidazolium in germ-tube formation was 10 µM in 1% PDB and fusion media. CAT induction was more sensitive than germ-tube formation to calmidazolium and had an MIC value of 5 µM. This shows that binding of intracellular Ca^2+^ to calmodulin is required for germ-tube formation and CAT induction, but as already observed with the other inhibitors, both developmental pathways display different requirements of intracellular Ca^2+^ levels.

### 3.4. Inhibition of Calcineurin Function Inhibits CAT Induction but Not Germ-Tube Formation

FK506 and cyclosporin A are indirect inhibitors of calcineurin function with different modes of action [36,54,55]. FK506 binds to FKBPs, while cyclosporin A binds to cyclophilins. Binding of either of these complexes to calcineurin blocks its function as phosphatase and prevents the dephosphorylation of downstream transcription factors. Here, we used concentration ranges of 0–4 µM of FK506 and 0–100 µM of cyclosporin A to determine the effect of calcineurin inhibition on germ-tube formation and CAT fusion. CAT fusion was inhibited by both agents; however, germ-tube formation was not (Table 1, Figure 2). The MIC value for the inhibition of CAT induction was 4 µM for FK506. Although less than 10% CATs were formed, we did not achieve complete inhibition of CAT induction, even at the highest tested concentrations of 100 µM cyclosporin A. Due to noticeable precipitation of the drug in the medium, it was not reasonable to raise the test concentration any further.

### 3.5. Calcium Sequestration through the Mitochondrial Calcium Uniporter Is Required for CAT Induction and Germination

In mammalian cells, RU360 inhibits the transfer of intracellular Ca^2+^ from the cytoplasm through the mitochondrial calcium uniporter onto the inner mitochondrial membrane. We tested the effects of 0.15–1 µM RU360 on germling morphogenesis (Table 1, Figure 2). Germ-tube formation in both 1% PDB and fusion medium exhibited a dose dependent inhibition, with MIC values of 0.75 µM and 1 µM in 1% PDB and fusion media, respectively. CAT fusion also exhibited dose-dependent inhibition, with an MIC value of 0.15 µM. Thus, our results indicate an inhibitory effect of RU360 on MCU in fungal cells. Germ-tube formation and CAT induction are hence dependent on intracellular Ca^2+^ sequestration to mitochondria, albeit at different concentrations of intracellular [Ca^2+^]. Influx of intracellular Ca^2+^ to mitochondria during CAT induction appears to occur at a higher concentration range of intracellular Ca^2+^ compared to germ-tube formation.

### 3.6. ER Calcium ATPase Inhibition by Thapsigargin Partially Inhibits CAT Induction but Not Germ-Tube Formation

Thapsigargin binds to the ER calcium ATPase and inhibits the sequestration of Ca^2+^ into the ER following a high concentration of [Ca^2+^] in the cytoplasm [39]. The test concentrations of 1–100 µM of thapsigargin resulted in less obvious dose-dependent effects in comparison to the other inhibitors (Table 1, Figure 2). Cell-symmetry breaking by germ-tube formation in 1% PDB alone was inhibited by more than 20 % at concentrations of 25 and 35 µM thapsigargin but not affected at higher concentrations. In fusion medium, 1–100 µM thapsigargin had only negligible inhibitory effects on germ-tube formation. CAT induction, on the other hand, was significantly inhibited by thapsigargin concentrations higher than 10 µM and decreasing to 40% at 100 µM. Notably, at (calculated) concentrations greater than 50 µM, the drug increasingly precipitated in the fusion medium, decreasing its actual concentration and bioavailability. Nevertheless, the data show that Ca^2+^ sequestration into the ER has a decisive role in cell-symmetry breaking by germ-tube and CAT formation, as well as CAT-mediated cell fusion.

## 4. Discussion

The goal of this study was to determine the role of Ca^2+^ signalling during germ-tube morphogenesis and CAT-mediated germling fusion in microconidia of *F. oxysporum* f. sp. *lycopersici*. Table 2 and Figure 3 summarize the results obtained with the different Ca^2+^-signalling inhibitors used in this investigation.

Experiments using the Ca^2+^ chelator BAPTA showed that extracellular calcium is not required for germ-tube formation. A previous study of *F. oxysporum* showed the induction of CAT fusion (~40%) in 1% PDB upon addition of 25 mM CaCl_2_ [13]. In the same study, no significant improvement in germination was observed upon addition of CaCl_2_. Similarly, Ca^2+^ signalling was shown to be dispensable for cell-symmetry breaking by germ-tube formation for spores of *Aspergillus nidulans*, *Magnaporthe oryzae* (formerly *M. grisea*) and *Curvularia trifolii* and proposed to be complemented by other pathways, such as the MAP kinase pathway and the PKA pathway [56,57,58].

Our results reveal that extracellular Ca^2+^ is required for CAT induction in *F. oxysporum*. The same finding was previously reported for *N. crassa* [59]. In this model fungus, the Ca^2+^ sensor mutant Δ*cse-1* exhibited a defective fusion phenotype. Deletion mutants of PIK-1 and NFH-2, two possible interacting partners of CSE-1, also showed similar phenotypic defects in CAT-mediated cell fusion as the Δ*cse-1* mutant [59]. Ca^2+^ signalling has been proposed to regulate the interaction between these three proteins, leading to exocytosis of the chemoattractant and/or receptor during hyphal fusion [59,60]. In *N. crassa*, PEF1, a protein involved in membrane plugging during injury, was shown to be dependent on extracellular Ca^2+^ for its membrane recruitment [61]. Extracellular Ca^2+^ chelation with EGTA also impaired septal plugging in response to wounds, independent of PEF1. Cell lysis involving transient membrane rupture and fusion is the key final step during CAT-mediated cell fusion. However, CAT formation itself was impaired in our study, highlighting the importance of extracellular Ca^2+^ for cell-cell communication during CAT induction, homing and, ultimately, CAT fusion. An oscillatory high-Ca^2+^ gradient was observed during polarized tip extension in *A. nidulans* [1]. On the other hand, it has been reported that the hyphae of *F. oxysporum* can grow in the absence of a tip-focussed Ca^2+^ gradient [3]. Evidently, different Ca^2+^ signatures could be involved in different types of tip growth. Hence, our results suggest that even though germ-tube protrusion involves polarized growth, it differs from polarized growth exhibited by CATs in terms of Ca^2+^ requirement. We previously showed that the negatively chemotropic growth behaviour of germ tubes is mainly driven by the CDC42 GTPase, whereas positive chemotropic growth behaviour of CATs requires the activity of RAC1 [62]. Similarly, different Ca^2+^-signalling pathways feeding into different downstream processes may add another layer of regulation to achieve the generation of functionally distinct cell protrusions from the same cell. While CATs can employ extracellular calcium as a chemotropic cue for CAT homing, germ tubes remain unresponsive. Therefore, selective Ca^2+^ inhibition with BAPTA presents a useful tool for discrimination of the signalling mechanisms required for germ-tube protrusion from those involved in self-signalling during CAT-mediated cell fusion.

Using the L-type Ca^2+^ channel blocker verapamil, both-germ tube and CAT formation were inhibited. Nevertheless, CAT formation was again more sensitive to this inhibitor, as previously seen with other inhibitors of cellular key processes, such as GTPases or microtubules [62,63]. In *S. cerevisiae*, verapamil was shown to partially inhibit Cch1-Mid1 activity [30]. Inhibition of hyphal development, adhesion and gastrointestinal-tract colonization upon treatment with verapamil was reported in *Candida albicans* [31]. Inhibition of appressorium formation but not conidial germination with 500 µM verapamil was reported in *M. grisea* (now *M. oryzae*) [57]. Amiodarone, an antiarrhythmic drug with a proposed activity in opening Ca^2+^ channels in the plasma membrane, leading to loss of intracellular Ca^2+^, exhibited antifungal activity against *F. oxysporum* [64]. Ca^2+^ channels mediate the flow of free Ca^2+^ along a concentration gradient into the cytoplasm, either from the external medium or from intracellular Ca^2+^ stores. Homologues of Cch1 and Mid1, two components of HACS, are present in filamentous fungi [2,43] and yeast and have been shown to play similar roles in *N. crassa* and *Giberella zeae* (the teleomorph of *F. graminearum*) [23,65,66]. Other mechanisms complementing the absence of CCH1, such as LACS component Fig1 have been reported [23]. Several studies in *N. crassa* and *S. cerevisea* have also provided evidence for the presence of Ca^2+^ channels other than cch1-mid1 in the plasma membrane [27,67,68]. Additionally, localization of cch1 has not been performed in fungi; the associated protein Mid 1 has been shown to be localized in ER in *B. cinerea* [53]. Unless cch1 is localized in the fungal plasma membrane, the effects observed with verapamil cannot be validated. However, pulsatile increases in cytosolic-free Ca^2+^ ([Ca^2+^]_c_) were observed in growing hyphal tips of the wild type and *FoCCH1* and *FoMID1* mutants in the *F. oxysporum* strain O-685, which is different from the tomato pathogen *F. oxysporum* strain 4287 used in this study, although the pattern differed between each of the mutants and the wild type [3]. A similar study in *F.graminearum* reported variations in Ca^2+^ signatures influenced by external calcium and growth defects of CCH1, MID1 and FIG1 mutants [69]. Ca^2+^ accumulation through CCH1 and MID1 in the plasma membrane was found to be important for thigmotropic tip growth in hyphae of *C. albicans*, although it was later shown to act as a signal enhancer for directional growth, with the possibility of additional essential signalling components [70,71]. Our findings support the function of HACS involving the L-type Ca^2+^ channel in the induction of germ-tube formation and in CAT fusion in *F. oxysporum*, especially when the availability of Ca^2+^ in the growth medium is low, such as in 1% PDB with or without NaNO_3_.

Calmidazolium, an inhibitor of calmodulin, was found to inhibit both germ-tube formation and CAT fusion. CAT formation was again more sensitive to this agent. Deletion of the gene encoding the calmodulin-binding protein HAM-3 was shown to be required for cell fusion [72], and loss of the Ca^2+^-binding protein HAM-10 was found to lead to fusion defects [73,74]. We also found that inhibition of the mitochondrial Ca^2+^ uniporter with RU360 inhibited both germ-tube formation and CAT fusion. CAT fusion was also more sensitive to this agent. This shows that mitochondrial Ca^2+^ stores are important for germ-tube formation and CAT formation although at different levels of sensitivity. Homologues of MCU have been identified in pathogenic filamentous fungi, including *Fusarium* spp. [47]. RU360 was found to alter the Ca^2+^ signature of transient increases in [Ca^2+^]_c_ during staurosporine-induced cell death in *N. crassa* [27,48]. On the other hand, inhibition of the ER Ca^2+^ ATPase by thapsigargin showed partial inhibitory effects on germination in 1% PDB or CAT induction in fusion medium. Germ-tube formation in fusion media was unaffected by inhibition of ER Ca^2+^ ATPase with thapsigargin. Our results with thapsigargin indicate that germ-tube formation could take place in the absence of Ca^2+^ uptake to the ER, albeit with reduced efficiency. A possible explanation is that the transient rise of cytoplasmic Ca^2+^ quickly falls back to resting levels due to alternative mechanisms involving other Ca^2+^-storage organelles, such as mitochondria or vacuoles, or by export to the extracellular medium. Germ-tube formation progresses since it is less sensitive to these changes, whereas CAT induction is inhibited, as it is dependent on Ca^2+^ uptake to the ER.

The HACS system involving CCH1 and MID1 has been shown to be sensitive to calcineurin and is therefore expected to have a functional role downstream of Ca^2+^ influx through these plasma membrane channels [75]. The calcineurin pathway is not essential for normal cell survival but is required for stress responses [76,77]. Our results indicate that inhibition of calcineurin by FK506-FKBP or cyclosporinA-cyclophilin inhibits CAT induction but not germ-tube formation in *F. oxysporum*. FK506 forms a complex with FKBPs, which, in turn, binds to the regulatory subunit B of calcineurin to inhibit its function [36]. Cyclosporin A forms a complex with cyclophilins and inhibits its action by binding to the same regulatory subunit of calcineurin [55]. Even though the inhibition of calcineurin by FK506 and cyclosporin A yielded the same result, where CAT induction alone was inhibited, FK506 is a more potent inhibitor of calcineurin, as previously reported [33,78]. In our study, CAT induction was found to be inhibited at lower concentrations of verapamil, calmidazolium and RU360 (Table 2) than germ-tube formation and therefore appeared to be much more sensitive to these agents. This further supports our notion that even though germ-tube formation and CAT induction both involve Ca^2+^ signalling, they are differentially regulated by Ca^2+^ channels, calmodulin and mitochondrial Ca^2+^ uniporters.

CATs were more sensitive to all pharmacological agents tested, which inhibited the process at much lower concentrations compared to germ-tube formation. Even though both cell protrusions do occur in the same medium (1% PDB + 25 mM NaNO_3_), germ-tube formation and CAT fusion occur at different time points during germling development [13]. Germ-tube formation is an early event in *F. oxysporum*, commencing ~2 h after incubation starts, whereas CAT fusion begins ~7 h after incubation starts. Our results also indicate the dependency of CAT fusion on a higher extracellular Ca^2+^ concentration, followed by its intracellular sequestration to mitochondria, as well as more Ca^2+^ binding to calmodulin when compared to germ-tube formation. Germ-tube formation, followed by continued germ-tube protrusion, increases the cell volume much more significantly than the protrusion of expansion-limited CATs. This might lead to further release of Ca^2+^ to the medium or alter the ratio of intracellular and extracellular Ca^2+^ levels. Thus, high extracellular Ca^2+^ availability following germ-tube formation might be a key prerequisite for CAT induction to occur in *Fol*. This is different from other filamentous fungi, such as *N. crassa*, in which CATs and germ tubes can be simultaneously formed by one and the same cell in Vogel’s media without other supplements.

## 5. Conclusions

Overall, our results provide clear evidence for a significant role of Ca^2+^ signalling in the initial morphogenesis of CATs and germ tubes. At the same time, our data strongly suggest a differential regulation of CAT induction and germ-tube formation by the Ca^2+^-signalling machinery. Our study provides further evidence of Ca^2+^ acting as a self-signal for CAT homing and fusion. Several features of cellular Ca^2+^ signals support this notion, including (1) their pulsatile nature in growing tips (fitting with the pulsatile ping-pong mechanism of SO and MAK-2 proteins to *N. crassa* tips [79]); (2) their ability to translate as elementary (localised and specific action) and global (waves and spikes) signals, which greatly facilitates signal translation across the intercellular space between sender and receiver [12]; and (3) their fast propagation dynamics, which is related to; (4) the fact that Ca^2+^ ions exist and function both intracellularly and extracellularly. The Ca^2+^ modulators used in this study were originally developed for use in mammals. As a result, their modes of action have been characterized in depth in mammalian cells. Hence, caution must be taken when interpreting the results as potential drug targets in *F. oxysporum* and other fungi until the specificity/selectivity for their fungal targets is confirmed. However, our studies highlight the utility of these pharmacological agents in studying calcium signalling in this important plant and human pathogen. This includes, for instance, the effect of verapamil, calmidazolium and RU360 in the inhibition of germ-tube formation, which presents important results for the development of novel antifungal drugs. Particularly with respect to increasing reports of acquired antifungal resistance of microbial pathogens, in future studies, it is crucial to identify fungus-specific targets in the universal Ca^2+^-signalling machinery of eukaryotes.

## Figures and Tables

**Figure 1 jof-08-00090-f001:**
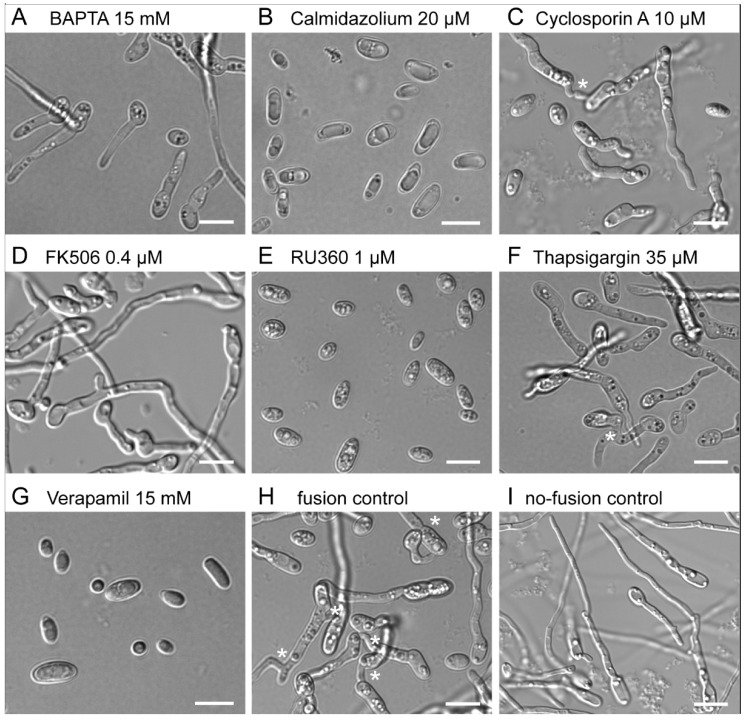
Effect of calcium-signalling inhibitors on germination and CAT-mediated cell fusion. (**A**) Chelation of extracellular Ca^2+^ with BAPTA inhibits the formation of CATs and thus blocks cell fusion very effectively. Germ-tube formation and elongation are unaffected. (**B**) Calmidazolium, which also blocks calmodulin/calcineurin signalling, already affects germination, germ-tube elongation and CAT-mediated cell fusion at low concentrations. Nevertheless, CAT formation is much more sensitive compared to germ-tube formation. (**C**) Cyclosporin A, on the other hand, which also acts on calcineurin signalling, has a stronger inhibitory effect on CAT-mediated cell fusion. (**D**) Inactivation of calcineurin signalling with FK506 has a strong inhibitory effect on CAT formation but not on germ-tube development. (**E**) Inhibition of mitochondrial calcium uniporters with RU360 results in a very similar phenotype compared to verapamil, though at much lower effective concentrations. (**F**) Inhibition of Ca^2+^ influx into the ER with thapsigargin has only a weak but specific effect on CAT formation. (**G**) Inactivation of Ca^2+^ plasma-membrane channels with verapamil supresses germination altogether. (**H**) In the positive control (1% PDB + NaNO_3_ fusion medium), germination, germ-tube elongation and germling fusion (*) occur with high frequencies, whereas (**I**) in the absence of NaNO_3_ (1% PDB only), CAT formation (and hence germling fusion) cannot take place, and only elongated germ tubes are produced. Scale bars, 10 µm.

**Figure 2 jof-08-00090-f002:**
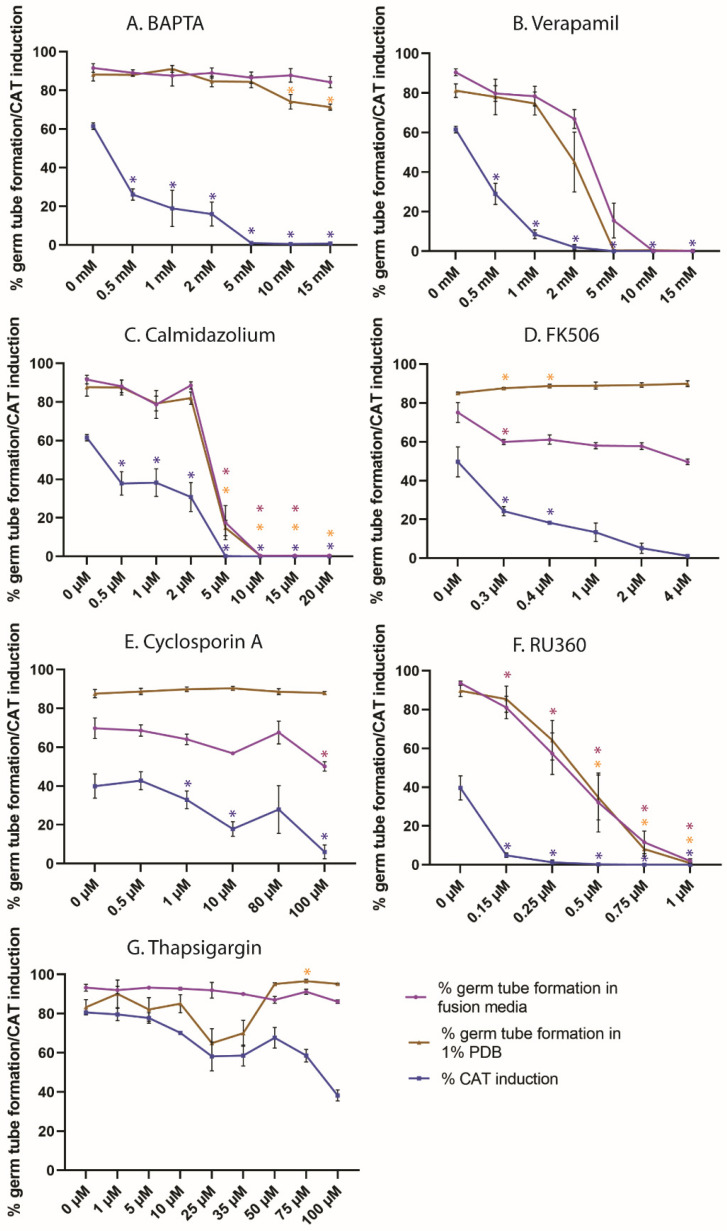
Effect of calcium-signalling inhibitors on germination and CAT-mediated cell fusion. (**A**) BAPTA, a chelator of extracellular Ca^2+^, inhibits the formation of CATs. Germ-tube formation and elongation remain unaffected. (**B**) Verapamil, an inhibitor of Ca^2+^ channels on the plasma membrane, inhibits CAT formation and germ-tube formation. (**C**) Calmidazolium, blocking calmodulin/calcineurin signalling, inhibits germ-tube formation and CAT formation. (**D**) FK506, an inhibitor of calcineurin signalling, inhibits CAT formation but not germ-tube development. (**E**) Cyclosporin A, an inhibitor of calcineurin, inhibits CAT induction but not germ-tube formation. (**F**) RU360, an inhibitor of mitochondrial calcium uniporters, inhibits germ-tube formation and CAT formation. (**G**) Thapsigargin, an inhibitor of ER Ca^2+^ ATPases, has a strong inhibitory effect on CAT formation and cell fusion but not germ-tube formation. *p*-values ≤ 0.05, indicating significant differences between individual drug concentrations in comparison to corresponding non-treated controls, are marked with an asterisk (*).

**Figure 3 jof-08-00090-f003:**
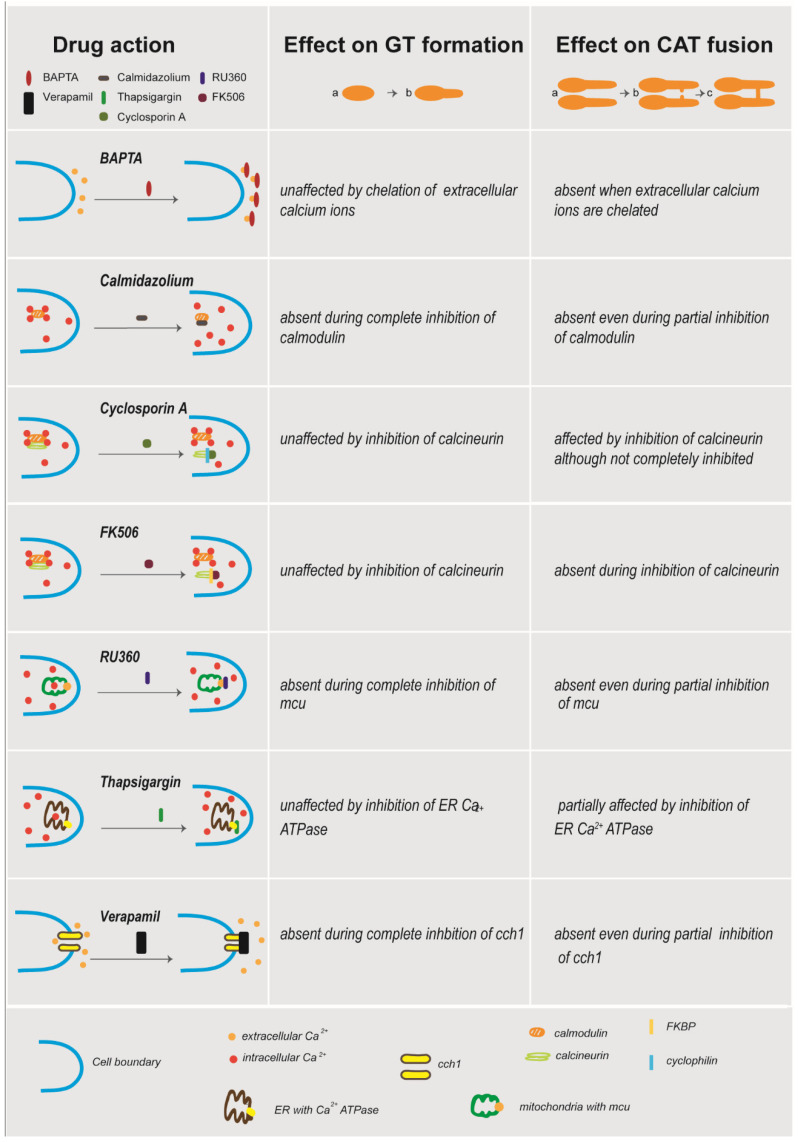
Schematic overview of the drug’s mode of action. The mode of action of each pharmacological agent in the fungal cell are correlated with the effect of each treatment on germ-tube (GT) formation and CAT formation.

**Table 1 jof-08-00090-t001:** Pharmacological agents used to inhibit Ca^2+^ signalling in *F. oxysporum*.

Name	Solvent	Source	Concentration
BAPTA	Water	Invitrogen Life Technologies	0–15 mM
Calmidazolium	DMSO	Acros Organics	0–20 µM
Cyclosporin A	DMSO	InvivoGen	0–100 µM
FK506	DMSO	InvivoGen	0–20 µM
RU360	Water	Calbiochem	0–1 µM
Thapsigargin	DMSO	Sigma Aldrich	0–100 µM
Verapamil	Water	Sigma Aldrich	0–15 mM

**Table 2 jof-08-00090-t002:** Pharmacological inhibition of germ-tube formation and CAT fusion. This table summarizes the effect of each pharmacological agent on germ-tube formation and CAT induction in *F. oxysporum*. The MIC values are derived from the corresponding graphs shown in Figure 2.

Pharmacological Agent	Mode of Action in Mammalian Cells	MIC for Germ-Tube Formation		MIC for CAT Induction
		1% PDB	Fusion Media	
BAPTA	extracellular Ca^2+^ ion chelation	no inhibition	no inhibition	5 mM
Calmidazolium	inhibits calmodulin	10 µM	10 µM	5 µM
Cyclosporin A	inhibits calcineurin	no inhibition	no inhibition	>100 µM
FK506	inhibits calcineurin	no inhibition	no inhibition	4 µM
RU360	inihibits mitochondrial Ca^2+^ uniporter	0.75 µM	1 µM	0.15 µM
Thapsigargin	inhibits ER Ca^2+^ ion sequestration	no inhibition	no inhibition	no full inhibition
Verapamil	inhibits Cch1	5 mM	10 mM	2 mM

## Data Availability

The data presented in this study are available on request from the corresponding author. The data are not publicly available due to the considerable number and large file size of the imaging data sets, as well as the requirement for specific image analysis software.

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
