# Peer review of "Ca2+ Signalling Differentially Regulates Germ-Tube Formation and Cell Fusion in Fusarium oxysporum"

_jof, 2022, doi:10.3390/jof8010090_

Round 1

Reviewer 1 Report

  1. There are too many statements in the abstract to summarize the background, and there are few statements about the research results. It is recommended to modify.
  2. The introduction part lists too much content, and I can not find the key points.The introduction part lacks logic, and the background related to this research is relatively small, which makes the proposal of the research too pale, especially the last paragraph of the proposal to summarize the content and significance of the key research.
  3. Materials and methods should be as detailed as possible so that future researchers can repeat the experiment and explain the reason for determining the index.
  4. The overall amount of data in the result part is too small, and there are too many language descriptions in the result part, and there is a lack of more relevant charts to support the conclusion. It is recommended to further supplement related experiments to illustrate the research.
  5. Figure 1 suggests to modify the scale representation in the picture. The scale line and the number are best combined, and the overall label can be placed below.
  6. The logic of the discussion needs to be strengthened. What needs to be discussed is the content related to your own results, rather than simply listing and quoting a large number of references.
  7. The conclusion is too long, and the conclusion needs to explain that it is the result of your personal research. It is recommended that the relevant part be added to the discussion, and the conclusion part should be condensed and deleted.
  8. The format of the reference is wrong, it is recommended to modify it according to the correct template.

Author Response

Please see attachment for response to reviewer 1

Reviewer 2 Report

The present work entitled “Ca2+ signaling differentially regulates germ tube formation and cell fusion in Fusarium oxysporum” has represented the comprehensive role and regulation of Ca2+ signalling components on germination and conidial anastomosis tubes (CAT) fusion for an important plant pathogen Fusarium oxysporum.

Overall the manuscript is well written, and the results are nicely correlated, I do not have any major concerns with the manuscript. However, I would like to suggest authors to include a few suggestions mentioned below:

Please include the statistical analysis in “Figure 2. Effect of calcium signalling inhibitors on germination and CAT-mediated cell fusion”. As, without statistical assessment, we cannot infer the significance of the results obtained.

There are several unnecessary citations, that resulted in the large no of citations i.e. 80, authors can reduce the inappropriate citation wherever possible.

Please expand the abbreviation used in first place, for example in line 51, 63 or wherever needed in the entire manuscript.

In line 116 “there are very few studies on the role of Ca2+ signaling in this important species.” This sentence should be supported with a citation that document the role of Ca2+ signalling in this regard.

In line no 125, citation 53 should not be placed in the bracket.

In line no 182 Please remove RU360 from “…results a very similar phenotype compared to Verapamil and RU360, however,…”

At some places author have used “signaling” while at some “signalling”. Authors should maintain a uniformity throughout the manuscript.

Author should critically survey the entire manuscript for similar errors to further improve the readability of the manuscript.

Author Response

Please see attachment response to reviewer 2

Reviewer 3 Report

Comment 1: Write details materials and methods section. Give separation section like results.

Comment 2: In figure 2, include information about statistical information like replicates and SD/SE

Comment 3: Please avoid two sentences as one paragraph throughout the manuscript.

Comment 4: Rewrite the discussion section, looks like too lengthy.

Comment 5: Write conclusion as a single paragraph.

Author Response

Comment 1: Write details materials and methods section. Give separation section like results.

Thanks for flagging this up. The ‘Materials and Methods’ section has been edited to include separate subheadings. Additional details have been included regarding the number of analysed images per experiment as well as the statistical analysis. The corresponding section from lines 162-173 now reads:“The formation of germ tubes and/or CATs were determined 12 hours post incubation (hpi) using simple DIC light microscopy and subsequent image quantification as detailed in Kurian et al., (2018). A total of 360 imaging files with each one comprising a 4x4 image array were collected for this study. From this collection of 5000 images, a minimum of 20 images were selected to morphologically evaluate at least 300 cells per test condition per experiment. Results were assembled from a minimum of three experimental repeats for each drug concentration tested and the mean ± SEM was plotted. …”.

Comment 2: In figure 2, include information about statistical information like replicates and SD/SE

This was already suggested by reviewer 2. In response, we have included the statistical analysis details in the Materials and Methods section and in Figure 2.

Comment 3: Please avoid two sentences as one paragraph throughout the manuscript.

Thanks for alerting us to this. We now combined sections in lines 107, 129, 396 and 424

Comment 4: Rewrite the discussion section, looks like too lengthy.

We went through the discussion section again and removed unnecessary citations (also in response to the comment made by reviewer 2). Other than that, we did not remove any text as we think it is all justified. Nevertheless, we did move certain lines from discussion to the conclusion which improves the readability of these final section.

Comment 5: Write conclusion as a single paragraph.

We prefer to keep both sections separate in order to clearly differentiate the more speculative outlook nature of the conclusion from the more results based discussion.

Round 2

Reviewer 1 Report

  1. The manuscript selected only 7 calcium signal inhibitors to determine spore germination and CAT-mediated cell fusion. The amount of data is too small and there is not enough evidence to prove the conclusion.
  2. The amount of data in this manuscript is too small to be published in journal of fungi as a research paper. The only valid datasof this manuscript are Figure 1 and Figure 2. Figure 3 is a summary of other people’s research results. Table 1 lists the names of different inhibitors and related information. Table 2 lists the content of Figure 2 which is summarized.
  3. The Ca2+ concentration produced by Fusarium oxysporumafter using 7 calcium signal inhibitors needs to be supplemented.
  4. After using an exogenous calcium signal inhibitor, it is necessary to increase the control blank with the same amount of Ca2+.
  5. The reference format of Figure 3 is inconsistent with the reference format of the full text. There are similar problems in line 285. Lines 248-254 (NaNO3is required for CAT-mediated germling fusion to occur in oxysporum. Consequently, CAT induction and fusion were only analyzed in the NaNO3 supplemented medium because the process is inhibited in the 1% PDB control condition. We have reported the absence of CAT fusion as an inhibition of the first stage in CAT-mediated cell fusion-which is CAT induction/CAT formation-because we did not find any condition involving the tested pharmacological agents where CATs were being formed but were unable to fuse.) have no references support.
  6. PDB control should be added in Figure 1.

Author Response

Comment 1: The manuscript selected only 7 calcium signal inhibitors to determine spore germination and CAT-mediated cell fusion. The amount of data is too small and there is not enough evidence to prove the conclusion.

Using seven calcium signal inhibitors helped us to unravel key features of calcium signalling on spore germination and CAT mediated cell fusion in a very comprehensive way. Even though it looks like a short study, extensive effort has been invested for collecting the images and manual quantitative analyses. We collected 20 images per drug concentration per experiment for each drug in this study. At leat three experimental repeats for each drug concentration have been perormed. In total, we evaluated more than 5000 images (20 x 43 different concentrations x 3 different repeats x 2 sets of images for different processes) to achieve the presented quantifications and insights. We are confident that the obtained results will be useful to other researchers that investigate calcium signalling in this important pathogenic fungus. Furthermore, so far, only two other studies have been published on calcium signalling in F. oxysporum f.sp. lycopersici (Hou et al.,2020, Kim et al., 2015). Therefore, we feel our contribution is valuable addition.

Comment 2: The amount of data in this manuscript is too small to be published in journal of fungi as a research paper. The only valid data of this manuscript are Figure 1 and Figure 2. Figure 3 is a summary of other people’s research results. Table 1 lists the names of different inhibitors and related information. Table 2 lists the content of Figure 2 which is summarized.

As outlined in response to the first comment we see value in our study. We furthermore do not agree that Figure 3 is a mere summary of research by others. Our results are summarised in the columns two and three whereas the references given in column four only support the mode of action of each drug explained by the diagram in column one. However, this comment shows that including the references in column four can be misinterpreted by readers as proofs for the conclusions we made in our study. Hence, in the present edited version, we have removed the references in column 4 to avoid such confusion.

Comment 3: The Ca2+ concentration produced by Fusarium oxysporum after using 7 calcium signal inhibitors needs to be supplemented.

This will require a new detailed study where ratiometric dyes are being used which chelate calcium ions. Alternately, as ours is a study with pharmacological agents, using a fluorescent Ca2+ reporter dye in combination will require normalization experiments for each drug concentration used. This will be a tedious task requiring additional facilities for fluorescent reads form cells.

In our opinion, the results in the present study with pharmacological inhibitors summarizes nicely the effect of Ca2+ signalling on germ tube formation and CAT fusion in F. oxysporum and, hence, provide novel insight without measuring the intracellular Ca2+ concentrations.

We appreciate the suggestion that further experimentation can improve the scope of this study. Most scientific investigations tend to lead to more new questions than they were able to answer in the first place. After first submission of the manuscript both first authors have moved to new jobs in new institutes. The initiator and last author of the study, Prof. Nick D. Read, tragically passed away in March 2020 and his lab was discontinued. Therefore, we are, unfortunately, not in the position to conduct further experimentation and thus hope that our publication will enable colleagues to follow up on the topic.

Comment 4: After using an exogenous calcium signal inhibitor, it is necessary to increase the control blank with the same amount of Ca2+.

We disagree that this is an appropriate control. Increasing the Ca2+ concentration in the control every time a drug of certain concentration is added is not helpful as it will change the basal calcium level in the cell. This changes the cellular state so significantly that it is not comparable to the actual test condition.

Comment 5: The reference format of Figure 3 is inconsistent with the reference format of the full text. There are similar problems in line 285. Lines 248-254 (NaNO3is required for CAT-mediated germling fusion to occur in oxysporum. Consequently, CAT induction and fusion were only analyzed in the NaNO3 supplemented medium because the process is inhibited in the 1% PDB control condition. We have reported the absence of CAT fusion as an inhibition of the first stage in CAT-mediated cell fusion-which is CAT induction/CAT formation-because we did not find any condition involving the tested pharmacological agents where CATs were being formed but were unable to fuse.) have no references support.

Thanks for flagging this up. We already removed the references from Figure 3 in response to this reviewer’s comment 2. We do not think a reference is required for lines 248-254 as we are referring to the different conditions used in this study.

Comment 6: PDB control should be added in Figure 1.

Thanks for flagging this up. Done.